

# In-flight characterization of a compact airborne quantum cascade laser absorption spectrometer

Linda Ort[1], Lenard Lukas Röder[1], Uwe Parchatka[1], Rainer Königstedt[1], Daniel Crowley[2], Frank Kunz[3], Ralf Wittkowski[3], Jos Lelieveld[1,4], and Horst Fischer[1]

[1]Atmospheric Chemistry Department, Max Planck Institute for Chemistry, Mainz, 55128, Germany
[2]Department of Dynamics at Surfaces, Max Planck Institute for Multidisciplinary Sciences, Göttingen, 37077, Germany
[3]Mechanical Support Department, Max Planck Institute for Chemistry, Mainz, 55128, Germany
[4]Climate and Atmosphere Research Center, The Cyprus Institute, Nicosia, 1645, Cyprus

**Correspondence:** Linda Ort (linda.ort@mpic.de) and Horst Fischer (horst.fischer@mpic.de)

**Abstract.** Here, we report the development of a new Quantum cascade Laser infrared Absorption Spectroscopy (QLAS) instrument, the Airborne Tropospheric Tracer In-situ Laser Absorption spectrometer (ATTILA), for atmospheric trace gas measurements on board of the German High-Altitude Long-range Observatory (HALO) aircraft. Its small and light design makes it suitable for airborne measurements up to approximately 150 hPa of ambient pressure (13 - 14 km). The dual laser

instrument can measure several trace gases simultaneously in two 36.4-m-path astigmatic Herriott cells with a data acquisition frequency of 1 Hz. We describe the measurement method and the data acquisition of ATTILA and its in-flight performance by focusing on potential sources of influences on the signal, which we investigated with a dedicated test flight during which the instrument sampled from a constant source. We show that linear critical influences associated with challenging movement patterns can be corrected afterwards, while non-linear limitations can be minimized by appropriate calibration frequencies and

integrated time intervals. During the recent aircraft campaign CAFE-Brazil (Chemistry of the Atmosphere Field Experiment in Brazil) from December 2022 to January 2023, carbon monoxide (CO) measurements from ATTILA show a good agreement of a $R^2$ of 0.89 with simultaneous CO measurements from an established IR spectrometer for airborne measurements, the TRacer In-Situ Tdlas for Atmospheric Research (TRISTAR), on a 10 s time resolution. First dynamical characteristics and tracer distributions of CO and methane ($CH_4$) over the Amazon rainforest can be identified with ATTILA measurements with

a total measurement uncertainty of 10.1 % and 17.5 % and a data accuracy of 0.3 % and 5.5 % for a data acquisition frequency of 1 Hz for CO and $CH_4$, respectively.

## 1   Introduction

Laser spectroscopic measurement techniques have developed considerably in the past two decades, improving in their resolution, accuracy and stability for trace gas measurements in the atmosphere (Werle, 1998). These advantages, and many

more, make laser spectroscopy a highly efficient technique for measuring trace gases compared to classic chemical and chromatographic measurement techniques (Li et al., 2013). Tunable Diode Laser Absorption Spectroscopy (TDLAS) and the more efficient Quantum cascade Laser infrared Absorption Spectroscopy (QLAS) have been proven to be reliable measurement tech-



niques in numerous publications (Faist et al. (1994); Kormann et al. (2002); Fried and Richter (2006); McManus et al. (2010); Tuzson et al. (2020); Pal and Pradhan (2021) and citations within). Especially for airborne measurements, a lightweight and compact design that can withstand the extremely difficult measurement conditions caused by abrupt changes in acceleration, direction and height is very important (Richter et al. (2015); Catoire et al. (2017)). QLAS and TDLAS trace gas measurements have been proven to be very valuable for detecting and understanding small-scale atmospheric chemistry, radiative effects, transport processes and their overall impact on the Earth's climate (Fried et al., 2008). A reliable spectrometer for airborne measurements is the TRacer In-Situ Tdlas for Atmospheric Research (TRISTAR), which performed successfully in a large number of field campaigns and research flights measuring tropospheric and stratospheric carbon monoxide (CO), carbon dioxide ($CO_2$), methane ($CH_4$), nitrous oxide ($N_2O$) and formaldehyde (HCHO) (Wienhold et al. (1998); Schiller et al. (2008); Tadic et al. (2017); Tomsche et al. (2019)). Since the work of Tomsche et al. (2019), TRISTAR has been further improved which is described in the study of Röder et al. (2023).

Airborne measurements can generate high-resolution and small-scale data even in remote locations, for example, in the tropics, where high cloud coverage complicates satellite-based measurements (Reiche et al. (2016) and citations within). Nevertheless, besides these advantages, experimental challenges such as vibrations, weight and size limitations, pressure and temperature fluctuations during research flights, and complicated measurement modes for operation which has prompted further investigations for suitable instrumentation (Li et al., 2013).

In the following, we present the Airborne Tropospheric Tracer In-situ Laser Absorption spectrometer, short ATTILA, which has been specifically developed for airborne trace gas measurements on board of the High Altitude Long range Observatory (HALO) aircraft. It is equipped with two room temperature quantum cascade lasers that measure CO, $CH_4$ and $N_2O$ in a two-cell system, which is described in more detail in section 2. ATTILA conducted measurements during the CAFE-Brazil campaign in December 2022 and January 2023, which took place above the Amazon rainforest region. In the scope of the campaign, a test flight was performed to investigate instrument behaviour in the face of extreme airborne challenges in more detail. The doubling of simultaneous trace gas measurements of the well known TRISTAR spectrometer and the newly developed spectrometer ATTILA allows a comparison and quantification of the data quality as well as some first insights into the atmospheric composition and trace gas distribution over the Amazon rainforest.

## 2   The ATTILA instrument

ATTILA is a custom-built two-cell mid-infrared room-temperature quantum cascade laser absorption spectrometer constructed in 2019 at the Max Planck Institute for Chemistry to measure trace gases in the atmosphere on board of the research aircraft HALO. It is designed to further improve airborne measurements considering the limitations of weight and size for small aircraft. ATTILA is mounted in a 19-inch box and is small in size (48 x 27 x 55 $cm^3$) with a total weight of only 20.6 kg (excluding the pump). It holds two optical frames mounted in parallel inside of it. Figure 1 shows a picture of the instrument setup inside the box. The instrument is layered into two sections. On the top layer, the two optical frames, the flow systems and the tubing are mounted. In the bottom layer the electronic devices are located. Those two layers are separated by a 4,5-mm-thick aluminium




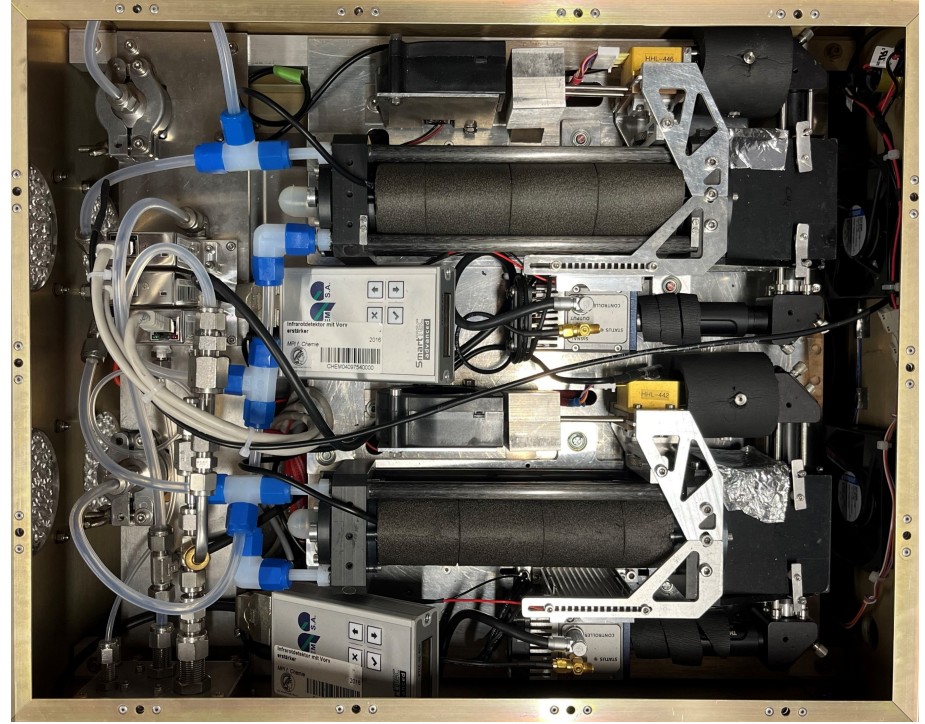

**Figure 1.** Picture of the top view of the measurement setup of ATTILA with the front of the instrument directed towards the right side. In the upper part of the picture the optical unit measuring $CH_4$ and $N_2O$ with the laser HHL-446 is located and in the lower part, CO and $N_2O$ can be measured with the HHL-442 laser.

plate and are ventilated by four fans located on the front of the instrument, which is in the picture on the right hand side. In the following the individual parts of the ATTILA instrument are described in more detail.

## 2.1 Optical and electrical setup

Figure 2 shows an overview of the schematic setup of the IR spectrometer. The two optical frames each consist of an astigmatic
Herriott cell (HC, Herriott and Schulte (1965); McManus et al. (1995)), a quantum cascade laser (QCL) and a mid-infrared detector, as well as mirrors and lenses directing the laser beam in the optical section. Multipass absorption cells are often used to provide long optical pathways in an optimized compact volume (Zahniser et al., 1995). The total air flow of $0.2 \, l \, min^{-1}$ is controlled by two compound mass flow controllers (MFC) (Bronkhorst, IQ+FLOW) separating the gas flow into the two cells with $0.1 \, l \, min^{-1}$ each. Furthermore, for calibration measurements another MFC leads $0.3 \, l \, min^{-1}$ of an external gas bottle into
the ambient tube, where $0.2 \, l \, min^{-1}$ are separated into the two HC again and $0.1 \, l \, min^{-1}$ prevent ambient air influence. The two QCLs (Alpes, Lasers, HHL-series, Lausanne, Switzerland) can measure CO ($2190.02 \, cm^{-1}$, Li et al. (2015)) and $N_2O$ ($2190.35 \, cm^{-1}$, Toth (2004)) in one cell and $CH_4$ ($1256.6 \, cm^{-1}$, Ba et al. (2013)) and $N_2O$ ($1255.42 \, cm^{-1}$, Toth (2004)) in the second cell. The $N_2O$ lines are mainly used to track the line centre position of the spectra throughout atmospheric mea-





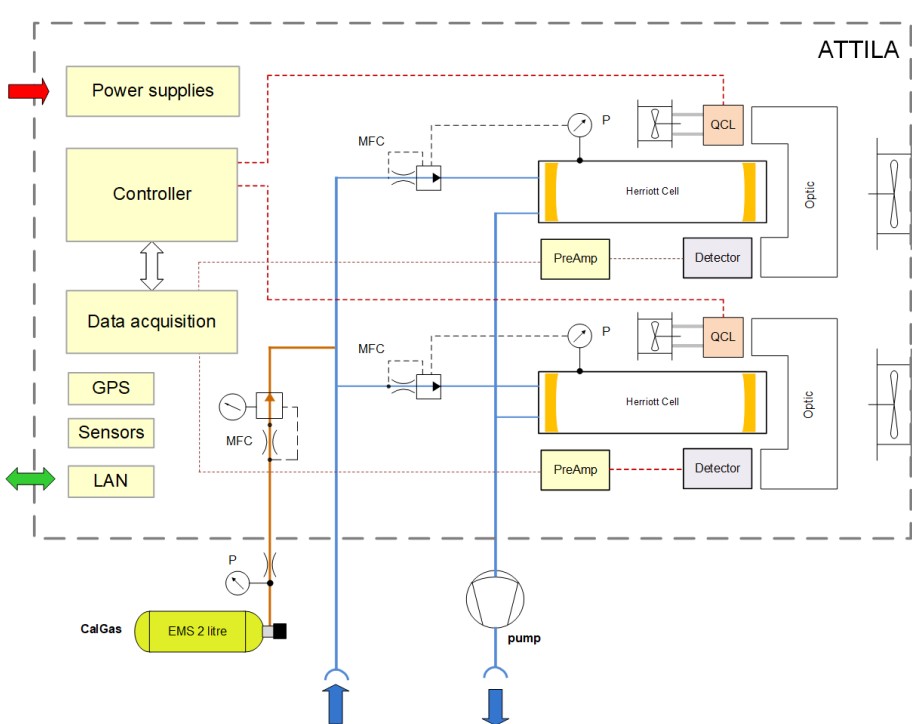

**Figure 2.** Schema of the measurement setup of ATTILA with the electrical and optical components.

surements due to its long lifetimes in the troposphere with only a small vertical gradient (Crutzen and Ehhalt, 1977). Both
lasers are functional for laser temperatures up to 60 °C but are cooled down to 26.5 °C for operation. Each laser is controlled
by a laser controller (Wavelength Electronics, WTC3243 & WTC3293), which adjusts the current and temperature of the laser.
An integrated Peltier junction and a NTC temperature sensor (model 10K4CG) together with the above-mentioned fans ensure
temperature stability. As the two optical frames are identical in their setup, the following description will focus only on one
frame, as shown schematically in Figure 3.

First, the laser beam passes two anti-reflection coated plano-convex lenses to compensate for the divergence and focus it onto
an adjustable plane mirror with an elliptical shape (Thorlabs, 12.7 mm (PFE05-P01)), which projects the laser beam on a right
angle prism mirror (Thorlabs, 12.5 mm (MRA12-P01)). Here, the laser beam is guided through an entrance window (Edmund
Optics, Techspec $CaF_2$ 20 mm) and a coupling hole of the front mirror onto the left side of the back mirror of the two opposing
astigmatic mirrors (Aerodyne, AMAC-36) of the HC. The correct adjustment of the HC is done with an optical visible adjust-
ment laser at 635 nm (Thorlabs, HLS635) to see and achieve the specifically needed Lissajous pattern (Zahniser et al., 1995)
on the two mirrors. This pattern is formed by the different horizontal and vertical radii of the curvature of $R_x = 246.0$ mm and
$R_y = 269.4$ mm, respectively, of the two concave mirrors with diameters of 3.8 cm. They need to be rotated exactly by 62.55°
to each other, considering their orientation, with the focus of the first entering beam in the middle of the cell. This alignment
results in a multipass state where the laser beam is reflected 182 times back and forth which covers a distance of 36.4 m in-



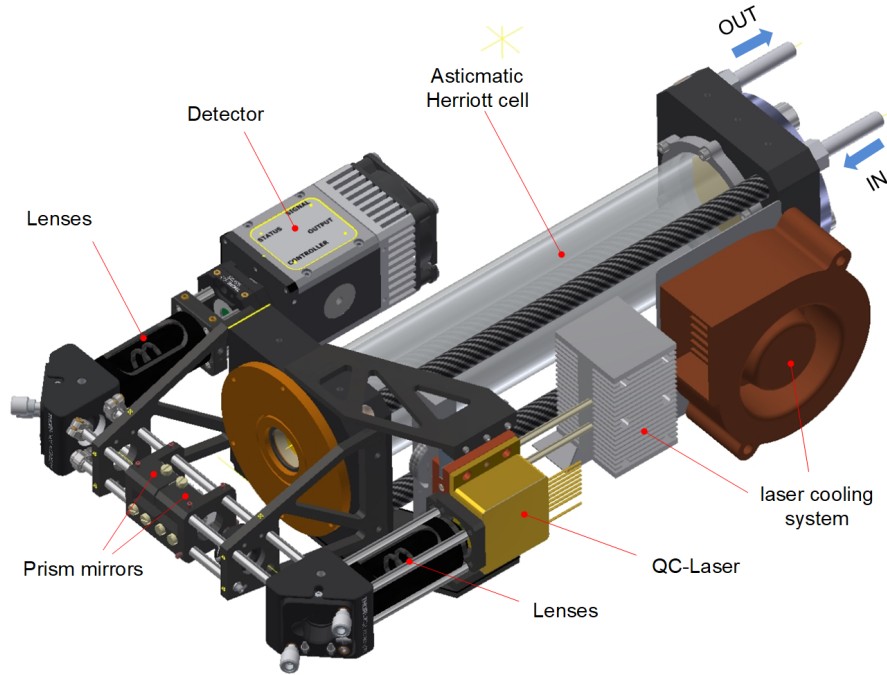

**Figure 3.** Sketch of the full optical setup for one cell including the astigmatic Herriott cell, the laser and detector as well as the directing mirrors.

side the compact HC with a length of 199.91 mm. With increasing path length the signal-to-noise ratio (SNR) increases until the loss of the many reflections becomes important (Werle and Slemr, 1991). With a volume of 250 ml, the astigmatic HC provides an effective SNR, compared to other HC types (Zahniser et al., 1995). Furthermore, between 3 to 10 $\mu$m the mirrors for the CO‑$N_2O$ cell are coated with silver, which provides a reflectivity of 99.2 % with a minimum of 99 % (Tafelmaier, Dünnschicht-Technik GmbH, Germany). For the same range but with a golden coating, the mirrors of the $CH_4$‑$N_2O$ cell have

a reflectivity of around 99 % (Tafelmaier, Dünnschicht-Technik GmbH, Germany). One potential limitation of the HC can be optical interference fringes produced by scattering in the cell which degrade the effective SNR and can disturb the molecular absorption lines through a similar free spectral range. More features about the HC can be found in McManus et al. (1995) and Zahniser et al. (1995).

While the laser beam passes the HC it gets partly absorbed by the trace gas of interest until it exits the cell through the entrance

hole again. Through two valves above the back mirror side of the HC the sampled air inside the cell can be exchanged. The input valve which is controlled by the above-mentioned MFCs brings either ambient or calibration gas into the cell on the front mirror side. Behind the output valve a pump (vacuubrand, Diaphragm pump - MD 1 VARIO-SP) is attached to reduce and keep the pressure inside the cells at 50 hPa continuously, which is detected by a pressure sensor (Dekont, VCC200) connected to the input valve. Furthermore, a PT-100 temperature sensor is mounted on top of the HC, detecting the temperature of the cell.

After exiting the HC through the front mirror hole the laser beam is directed with a prism and an adjustable elliptical mir-





ror through a single lens and focused onto a room-temperature infrared sensitive photovoltaic semiconductor detector (VIGO System S.A., PIP-DC-20M-F-M4). The beam slightly overshoots the detector entrance to compensate for potential beam misalignment due to vibrations.

For improved stability of the signal during vibrations and movements the optical frame is equipped with bracket reinforcements.
Furthermore, several black isolating covers on top of the laser pathways reduce temperature fluctuations and laser scattering.

## 2.2 Signal processing and analysis software

The characteristics of the signal are processed, controlled and read out by a single board controller (National Instruments, CompactRIO sbRIO-9627). The controller includes a real time processor, a Field Programmable Gate Array (FPGA), which
modulates the frequency on top of a voltage ramp including an offset to scan the targeted spectrum range. This is done by using the Wavelength Modulation Spectroscopy (WMS) technique, described in Fried and Richter (2006). The triangular ramp provides two spectra, an upwards- and downwards-directed spectrum of the measured signal. With a sinus-wave modulation frequency of 17.86 kHz, a ramp is generated with 4096 discrete steps and scanned within 28.7 ms. This provides a scanning frequency of approx. 35000 samples per second for each tracer of interest, after which the ramp is reduced by a quarter
resulting in a 1024-points spectrum. This digital command is converted into an analogue signal via a digital-analog-converter (DAC) and transferred through an analog output to the laser controller. Then, the laser controller can adjust the temperature and current of the infra-red laser to perform the wavelength scan of the signal to generate an absorption spectrum while passing the HC. After detecting the absorbed signal a pre-amplifier forwards the amplified signal through an analog input back into the sbRIO, where the signal gets digitized by an ADC again. Here, a digital lock-in amplifier multiplies and phase
shifts the signal with the doubled initial modulation frequency ($2f$) and to match the phase of the measured signal. The $2f$ harmonic signal provides a baseline near zero volts and a maximum at the centre of the line (Fried and Richter, 2006). This ensures a minimization of electronic noise and an in-dependency of the baseline of the spectra. Therefore, background calibrations and background subtraction are not necessary. Only a source with constant and known gas concentrations is needed to calculate the final gas mixing ratios. After the lock-in amplifier, a low-pass filter generates the DC signal with 1024
channels and an absorption spectrum which can be visualized in the corresponding software LabView (Laboratory Virtual Instrument Engineering Workbench). The signal processing methodology and setup has a resolution time of 1 Hz with a duty cycle of 100 % for each tracer of interest, thereby facilitating the real-time visualization of precise concentrations of trace gases present in the sampled air. Additionally, a linear regression scheme calculates real concentration values, $c_s$, with the slope of the correlation between the fit of the $2f$ absorption spectrum $S_s$ and the calibration spectrum $S_{cal}$ multiplied by the known
calibration gas concentration $c_{cal}$ as followes (Fried and Richter, 2006):

$$c_s = c_{cal} \cdot \mathrm{corr}(S_s, S_{cal}) \tag{1}$$

A huge advantage of the processing is the flexibility of the modulation and fitting settings. Every parameter can be digitally adjusted to achieve the best result. The current settings have been chosen after intensive laboratory test runs and have proven



themselves during the campaign.

In addition to the online signal processing, a specialized analysis software was developed using IGOR Pro (Wavemetrics) for comprehensive post-processing of the raw spectra. The software features automated filtering, exact positioning of the absorption lines and background subtraction, if necessary, while also facilitating a precise linear fit to a calibration spectra. Moreover, linear cell pressure variances as well as non-linear Lambert-Beer-effects can be corrected and overall precise mixing ratios of a species of interest can be determined. Furthermore, 1D and 2D visualization of the spectra with time can be used

to recapitulate and correct for drift behaviour or abrupt changes in the background induced by, e.g., temperature and pressure changes as well as the movements of the aircraft (Schiller et al., 2008). Further analysis and noise reduction opportunities have been discussed in Röder and Fischer (2022) and citations within.

### 2.3 Instrument setup on board of HALO

ATTILA is developed for aircraft measurements, specifically on board of the research aircraft HALO. During field campaigns,

it is mounted in a 19" measurement frame rack together with the Nitrogen Oxide Analyzer for HALO (NOAH), described by Tadic et al. (2020) and Nussbaumer et al. (2021). Both instruments sample continuously over a 6.28-m-long bypass from

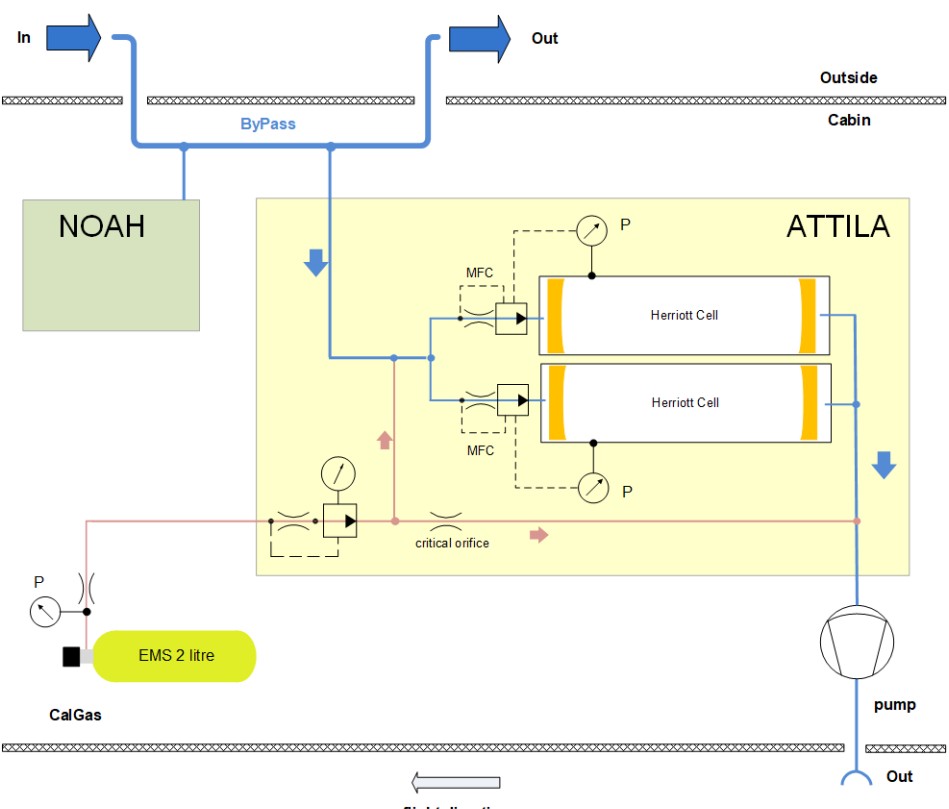

**Figure 4.** Flow chart of ATTILA setup on the HALO aircraft.





the top of the aircraft through a forward-facing stainless steel Trace Gas Inlet (TGI). The Teflon-coated tubing channels the ambient air through 1/2" PFA (perfluoro alkoxy alkane) tubes until it exits the aircraft again through a second TGI. After $4.57\,\mathrm{m}$ of this tubing, $0.2\,\mathrm{l\,min^{-1}}$ is led through a 1.27-m-long 1/4" tube into the ATTILA instrument, $0.16\,\mathrm{m}$ before the nitrogen oxide analyzer samples $3.0\,\mathrm{l\,min^{-1}}$ from the bypass. Figure 4 shows a sketch of the gas flow. As the ambient air enters
the spectrometer, the air is separated into the two HCs through two MFCs as described above.

For in-flight calibrations, a 2 l EMS gas bottle filled with compressed air is used. It is stored below the rack during the flight. The exact concentrations of the compressed air are characterized with a primary standardized gas bottle beforehand. During the calibration mode, one MFC leads the calibration gas with a flow of approximately $0.3\,\mathrm{l\,min^{-1}}$ into the other two MFCs, which separate the flow into the cells with $0.1\,\mathrm{l\,min^{-1}}$ each. Due to this overflow of $0.1\,\mathrm{l\,min^{-1}}$ between calibration and
ambient measurements, the calibration gas steams slightly into the ambient and exhaust lines which prevents these lines from influencing the measurements. A critical orifice is built inside the connecting tube towards the exhaust that the latter does not influence ambient measurements.

## 3   Instrument performance based on test flight

Within the CAFE-Brazil campaign from November 2022 to January 2023, 22 flights haven been performed with HALO, in-
cluding two test flights during the preparation phase at the DLR in Oberpfaffenhofen, Germany, 4 transit flights with a stop-over at Sal, Cape Verde and 16 research flights over the Amazon rainforest based in Manaus, Brazil. The first test flight was used to investigate the in-flight behaviour of ATTILA. Therefore, continuous calibration gas measurements have been performed during this test flight on the 22[nd] of November 2022 in the preparation phase of the CAFE-Brazil campaign. This test flight was specifically planned to investigate the behaviour of individual instruments in response to in-flight environmental changes like
temperature and pressure, as well as extreme aircraft movements and has already been used for instrument characterization in the studies of Hamryszczak et al. (2023) and Röder et al. (2023). Hence, various flight levels, different vertical velocities and accelerations as well as the variation of yaw, pitch and roll angles have been performed. At 10:30 UTC the aircraft departed. Before, the instrument was running for approx. two hours in the aircraft to warm up. The final approach of HALO for that test flight was at 14:23 UTC. Several instrument flight parameters are shown in Figures 5 and 6 and are discussed to determine the
systematic influences on the signal.

Starting with Figure 5, the 2D $2f$ raw spectra with the absorption lines of CO and $N_2O$ are shown in blue in the channels 200 and 800, respectively, over time. Additionally, the temperature of the HC $T_{HC}$, as well as the cabin pressure $p_{cab}$ of the aircraft measured at the rack is added as yellow and grey lines, respectively. Please note that the spectrum in Figure 5 is already processed with a pre-filter and the absorption lines are locked and shifted according to their line maximum. The background
signal has not been shifted. By focusing on the background signal between channel 300 and 700, thus between the absorption lines of CO and $N_2O$, interference on the signal can be addressed. The long-term drift behaviour of the spectra correlates well with the change of the temperature inside the HC. $T_{HC}$ changes during the test flight by about $6\,°C$ peak to peak. These changes are influenced by the cabin temperature which fluctuates in the same range even with the air conditioning control system on



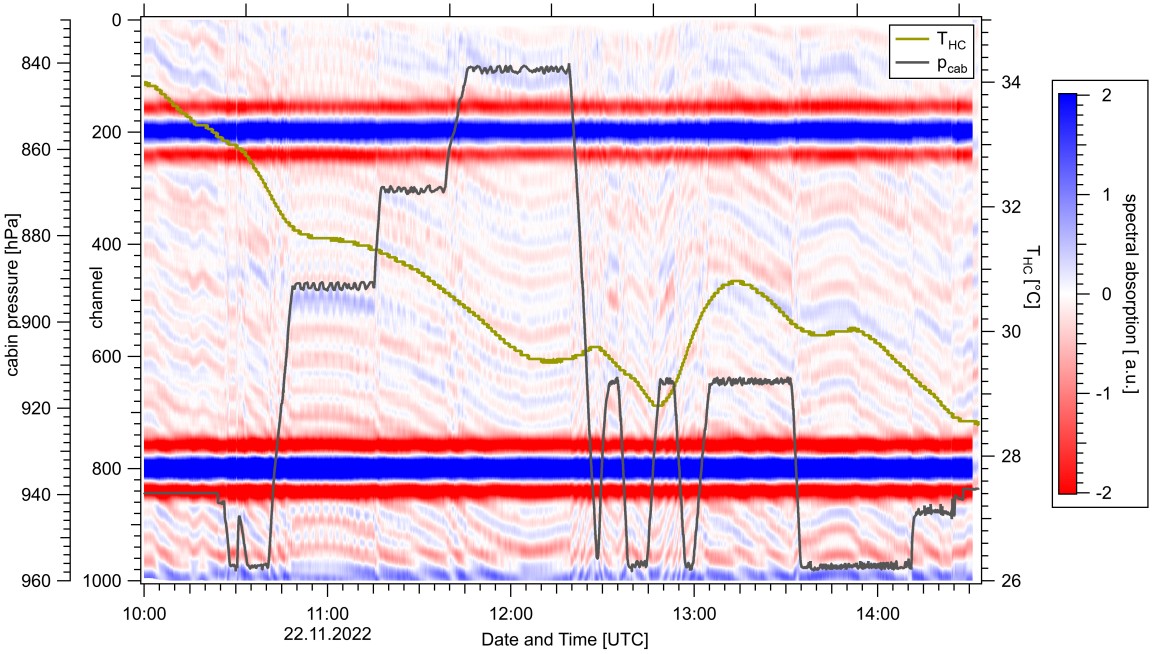

**Figure 5.** The 2D spectrum of the upwards-directed ramp signal of the CO‑N₂O‑laser showing the channels versus the time of the first test flight on the 22$^{\text{nd}}$ of November 2022 with the $N_2O$ absorption line in the bottom (channel 800) and the CO absorption line displayed in the top (channel 200) of the graph. The color scale is adjusted to values close to zero to show a stronger contrast of the background signal. Blue colors indicate positive values, hence absorption of the signal and red colors negative values of the $2f$ spectrum. The yellow line shows the temperature on top of the HC during the whole flight. In grey, the cabin pressure is displayed, measured at the ATTILA rack. The HALO aircraft was in the air from 10:30‑14:23 UTC.

board of the HALO aircraft. The post-processing of the signal includes a shift of the absorption lines by locking the signal on
its line maximum. With this, the drift behaviour of the signal can be corrected, which can be seen in Figure 5. Furthermore,
temperature changes have a small impact on the laser temperature and its operating point and therefore the intensity of the
signal. Looking at the laser temperature of the CO‑N₂O‑laser in Figure 6 in the lowermost panel, the temperature change cor-
relates well with the one from $T_{HC}$. A variation of temperature further influences the optical setup by expansion of the material
to induce etalon structures through interference. These interference fringes can influence the absorption signal amplitude via
non-linear constructive and destructive interference, which are much harder to identify. Slow changes in the fringe structure,
such as temperature changes mentioned above, can be approximated linearly and thus be counteracted by frequent calibrations.
Moreover, a change of pressure can have an influence on the signal. There are two influences known, which lead to different
assumptions. First, a change of ambient pressure and second a change in cabin pressure. First, when the ambient pressure
changes, the pumping has to adjust to the new pressure gradient to keep the flow stable. This is done by varying the voltage of
the pump from approx. 4 V up to its maximum of 10 V. The pump voltage, the HC pressure and the pressure gradient calcu-
lated from the ratio of the ambient to the cabin pressure are shown in Figure 6. Especially for fast altitude changes during, e.g.,



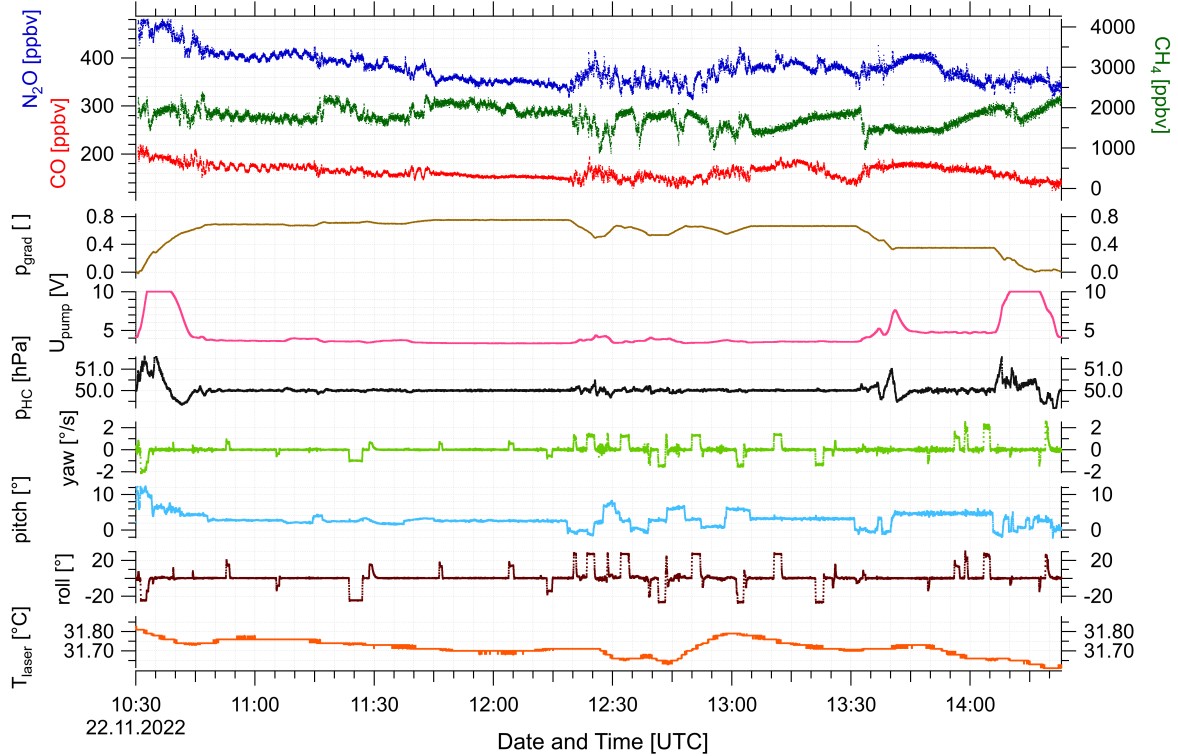

**Figure 6.** Compilation of instrument and flight parameters over time from the test flight on the $22^{nd}$ of November 2022. Shown are the laser temperature of the CO - $N_2O$ - laser ($T_{laser}$, orange line), the voltage of the pump ($U_{pump}$, pink line), the mean HC pressure calculated over both cells ($p_{HC}$, black line) and the roll (dark red line) and pitch (light blue) angles as well as the yaw angle rate per second (light green) of the HALO aircraft during that flight. As ocher line the pressure gradient calculated as the ratio of the ambient pressure and the cabin pressure as $p_{grad} = 1 - p_{ambient} / p_{cabin}$ is displayed. Note that the tracer concentrations of CO in red, $N_2O$ in blue and $CH_4$ in green in the top panel of the graph are not yet corrected by pressure and Lambert-Beer. A linear calibration fit has been used from only one calibration before the flight. Throughout the whole flight ATTILA was running on calibration mode sampling only from a gas bottle with $161 \pm 9$ ppbv of CO, $358 \pm 22$ ppbv of $N_2O$ and $2025 \pm 7$ ppbv of $CH_4$, including the accuracy of the bottle.

departure or landing, these disturbances are visible in the cell pressure, spectra and concentration. At constant flight levels and slow altitude changes, the pump can easily regulate the cell pressure to keep it constant at 50 hPa. Cell pressure instabilities are linear and hence can be corrected in the data processing.

Secondly, Figure 5 shows that the cabin pressure of the aircraft fluctuates even in constant flight levels of about 4 hPa. This influences the spectra and hence the measured concentration by inducing sine-structured short-term fluctuations. This can be seen especially between 10:48 and 11:15 UTC. Looking at the three different flight levels between 10:48 and 12:18 UTC, the cabin pressure fluctuations do not change but their influence on the signal does as can be seen in the mixing ratio in Figure 6. Hence, we see that pressure instability has an effect on the interference fringes. This effect is not linear and still needs to be





investigated.

In the first half of the test flight the aircraft mostly flies straight, which can be seen from the roll and pitch angles and the yaw angle rate displayed in Figure 6. Small aircraft movements only have minor influences on the signal. In the second half of the flight from 12:30 UTC on wards extreme aircraft movements involving abrupt changes to varying roll, pitch and yaw angles are carried out. These aircraft movements create disturbances in the signal through, e.g., centrifugal forces which can have an

effect on the optical setup and hence the signal. The mixing ratio in Figure 6 shows a strong impact of aircraft movements on the signal. However, these extreme aircraft movements are, if possible, avoided during measurement flights to provide as stable conditions as possible for airborne measurements. Nevertheless, another important factor causing small instrumental movements might be atmospheric turbulence which is not shown here.

In general, influences on the signal are mostly induced by temperature and pressure changes and aircraft movements (Fried and

Richter, 2006), which create linear and non-linear effects, as described above. As it is mostly a combination of variables that influence the signal, disentangling these effects of individual parameters is difficult. Nevertheless, temperature drift behaviour, pressure effects and short-term disturbances can be corrected by frequent calibration measurements and averaging (Richter et al., 2015). To investigate the appropriate calibration frequency, different calibration intervals have been tried out in the post-processing of the test flight, assuming that parts of the calibration gas measurements are calibrations and others are ambient

measurements. The tracer concentrations shown in Figure 6 are not yet drift-corrected. Their mean values are $163 \pm 14$ ppbv, $378 \pm 30$ ppbv and $1768 \pm 235$ ppbv for CO, $N_2O$ and $CH_4$, respectively. For the optimal drift correction different calibration intervals have been tried out in the post-processing of the test flight data. The drift-corrected mixing ratios for different calibration intervals for CO, $N_2O$ and $CH_4$ can be found in the supplemental materials in S1, S2 and S3, respectively. The higher the interval, the better drift influences can be corrected, but the less ambient measurements are done and the more calibration gas is

needed. The drift correction clearly depends on the positioning of the calibrations and the interference at that time. A measure for this is the reproducibility $R$. For this, a mean value of each calibration measurement is calculated and investigated for their accuracy and shift over time. Low $R$ indicates similar mean values of each calibration and hence small drift behaviour. $R$ varied from 0.71 to 8.54 % for different calibration frequency scenarios from every 85 minutes to every 2 minutes. After drift correction, the relative statistical error of the calibrations provides the mean precision $P$ which is 2.63 $\pm 0.47$ % for all scenarios.

For the choice of the appropriate calibration interval those different scenarios have been compared and an automated interval of 20 minutes was chosen as an optimal trade-off between drift counteraction, duty cycle and calibration gas consumption. The duration of the calibrations are 60 seconds, where the first 15 seconds are not taken into account in the analysis to ensure sufficient flushing time of the cell. With 20 minutes of calibration frequency the mean values of the corrected mixing ratios are $160 \pm 10$ ppbv, $357 \pm 14$ ppbv and $2029 \pm 197$ ppbv for CO, $N_2O$ and $CH_4$, respectively. Therefore, they are in the range of the

standard error of the calibration gas which are for CO $161 \pm 9$ ppbv, for $N_2O$ $358 \pm 22$ ppbv and for $CH_4$ $2025 \pm 7$ ppbv. This calibration setting was adopted for the rest of the campaign.



| Tracer | $P \pm \sigma$ [%] | $R \pm \sigma$ [%] | $MU$ [%] | $Acc$ [%] |
|---|---|---|---|---|
| CO (ATTILA) | $2.6 \pm 1.2$ | $9.8 \pm 3.9$ | 10.1 | 0.3 |
| N$_2$O (ATTILA) | $1.4 \pm 0.3$ | $9.1 \pm 4.2$ | 9.2 | 0.1 |
| CH$_4$ (ATTILA) | $4.2 \pm 1.5$ | $17 \pm 9.5$ | 17.5 | 5.5 |
| CO (TRISTAR) | $0.3 \pm 0.04$ | $3.5 \pm 1.9$ | 3.5 | 0.3 |
| N$_2$O (TRISTAR) | $0.2 \pm 0.04$ | $2.97 \pm 2.05$ | 3.0 | 0.1 |

**Table 1.** Mean precision $P$, reproducibility $R$ including their 1-$\sigma$, and the total measurement uncertainty $MU$ calculated using Eq. 2 for all research flights of the CAFE-Brazil campaign for the ATTILA and TRISTAR spectrometers. The accuracy $Acc$ of the secondary standard used as calibration gas during the flights is calculated using Eq. 3.

## 4 Instrument comparison for ambient data

The long-established TRISTAR instrument (e.g. Wienhold et al. (1998); Schiller et al. (2008); Tadic et al. (2017); Tomsche et al. (2019)) was deployed simultaneously to ATTILA within the CAFE-Brazil campaign, measuring also CO and N$_2$O. Both IR spectrometers operated throughout the whole campaign without any technical malfunctions. A comparison of these two IR spectrometers has been done to investigate the instrumental performance of the newly developed spectrometer. This is especially important for future planning, as ATTILA can take over CO measurements and TRISTAR could be modified to measure other trace gases like, e.g., formaldehyde due to its high stability and precision. Please note that the general measurement focus of the instrument is on CO, as the N$_2$O absorption line is only used to track the spectra on its right position, especially in high altitudes where CO concentrations are markedly lower. The CH$_4$ data will be shown in the next section.

The total in-flight measurement uncertainty $MU$ over the whole campaign is estimated by using $R$ and $P$, described in Section 3 and calculated as followed:

$$MU = \sqrt{P^2 + R^2} \tag{2}$$

Another measure for data comparison is the accuracy $Acc$, representing the standard error of the in-flight calibration gas. $Acc$ of the secondary standard used as calibration gas during the flights is calculated via

$$Acc = \sqrt{(R_F)^2 + (\frac{\sigma}{N})^2 + (\frac{F_{err}}{F_{mean}})^2} \tag{3}$$

including the reproducibility of the primary standard measurements $R_F$, the standard deviation divided by the number of points of the secondary standard measurements and the primary standard error over its mean content.

$MU$ is in the lower percent range for both instruments as shown in Table 1. In comparison to the performance of TRISTAR, ATTILA's $MU$ is increased by a factor of three. $Acc$ is for all tracers in the per-mille range, except for CH$_4$ which is at 5.5 %. For the comparison, a time resolution adjustment has been made. As an example, a section of the research flight number 14 on



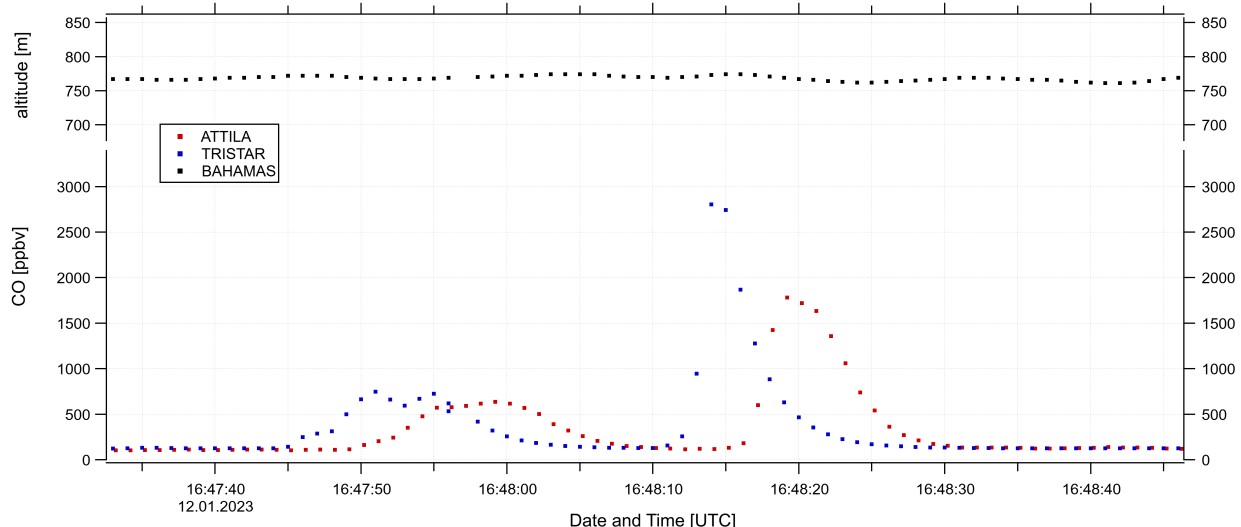

**Figure 7.** Timeline of CO mixing ratios from ATTILA in red and TRISTAR in blue measured during RF14 on the 12[th] of January 2023. In black, the GPS altitude from the BAHAMAS instrument (DLR, Oberpfaffenhofen, Germany) during the flight is displayed. All data is of 1 Hz resolution. The CO measurements have been fully processed and corrected by drift, pressure change and Lambert-Beer.

the 12[th] of January 2023 is shown in Figure 7 where extreme CO concentrations could be measured with both instruments, as the aircraft flew through several fresh biomass burning plumes at a low altitude in the boundary layer of about 760 m. In Figure

7 two smaller plumes are displayed with high concentrations above 1000 ppbv. There is a delay of five seconds visible between the two instruments, which is due to the differences in their flow rate and cell volumes. TRISTAR has a flow rate of $3 \, \mathrm{l \, min^{-1}}$ and hence exchanges the sampled air much faster then ATTILA. This time delay of 5 s has been corrected for further analysis of the ATTILA data. Another consequence of the time resolution difference is that TRISTAR measures higher concentrations in a short extreme event which ATTILA can only see smoothed out with its flow of $0.2 \, \mathrm{l \, min^{-1}}$. High frequency features below

5 s cannot be measured with ATTILA. This, specifically, shows the first peak at 16:47:45 UTC of the biomass burning event displayed in Figure 7. Nevertheless, these flow limitations are only a few seconds which is still a good time resolution for airborne measurements.

In Figure 8 the correlation of all CO measurements throughout the whole campaign from both IR spectrometers is displayed. Additionally, a least orthogonal distance fit has been performed, including their $MU$s (Table 1). The scale has been slightly

adjusted to focus on typical atmospheric concentrations instead of extreme events. Except for some outliers, which can be related to extreme aircraft movements and changes during some take-off and landings, the 1 Hz measurements agree quite well with a slope of $1.0022 \pm 0.0004$ and a $R^2$ of 0.83. The differences in the data sets are mainly due to their different instrumental noise. TRISTAR features a more robust design, a better insulation and temperature control through a heating plate, a stabilized and more extensive optical system and its operation can profit from many years of experience and further improvement of the

spectrometer throughout numerous campaigns (Tomsche et al. (2019), Tadic et al. (2017), Schiller et al. (2008) and citations




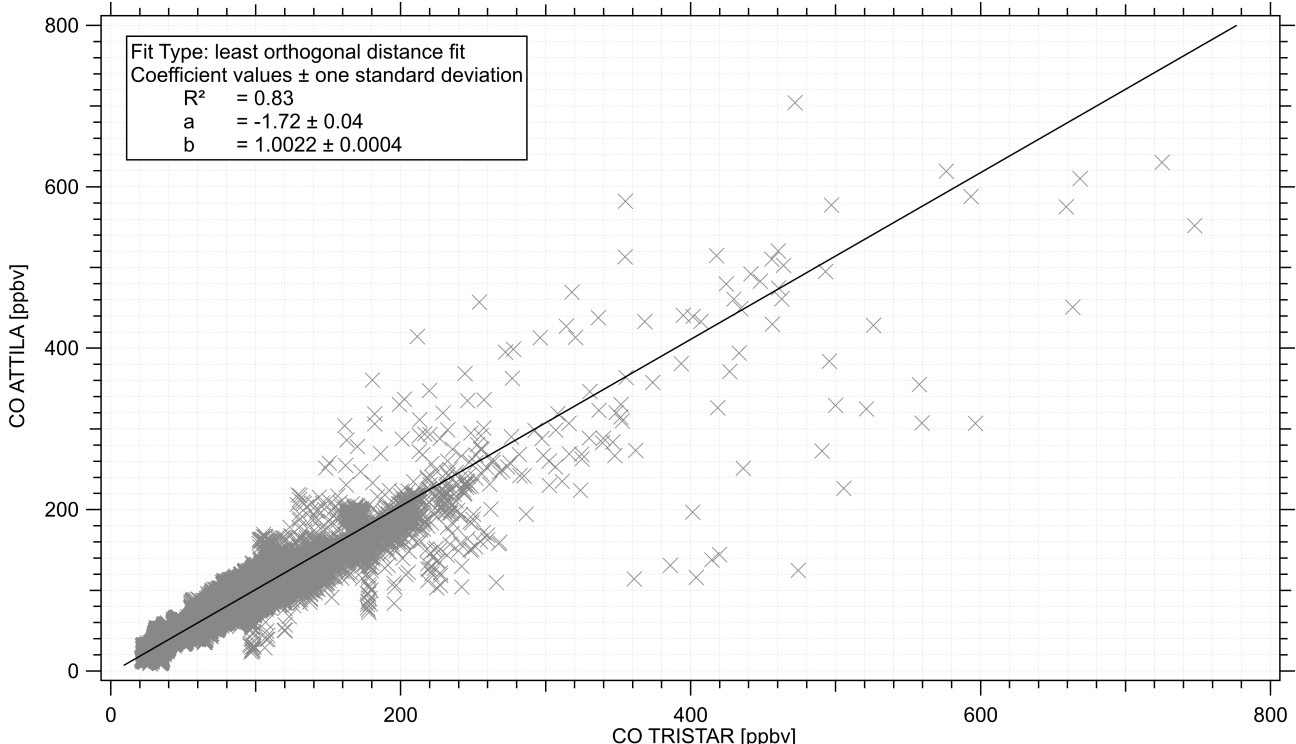

**Figure 8.** Correlation of all CO ambient measurements from ATTILA on the y-axis and from TRISTAR on the x-axis over the whole CAFE-Brazil campaign, excluding the two test flights. The data has been fully processed and corrected. A least orthogonal distance fit, displayed as a black line, has been performed, including the $MU$ values presented in Table 1. Outliers are mostly related to extreme aircraft movements during take-off and landings.

within). Therefore, fast atmospheric changes and unusual extreme events in the boundary layer with CO measurements higher than 600 ppbv can be better detected by the TRISTAR instrument. Nevertheless, an averaged time resolution of 10 seconds provides a $R^2$ of 0.89 with a slope of $0.9811 \pm 0.0003$, and is an adequate time resolution illuminating nearly all the instrumental noise. The correlation of the 10 s averaged data is displayed in Figure S4 in the supplemental material.

In Figure 9 the 1 Hz data has been binned into 500-m boxes from the whole campaign data set excluding the two test flights. In green and black the mean, median and standard deviation of ATTILA and TRISTAR, respectively, are shown. A few assumptions can be made. First of all, below 1 km several extreme events have been measured due to anthropogenic emissions and biomass burning events. As explained above, due to TRISTAR's higher time resolution and flow, short-term changes in extreme mixing ratios can be detected better with TRISTAR, leading to a higher variation of the standard deviation near the surface

compared to ATTILA CO measurements. Nevertheless, their mean values differ only by 7 ppbv. Furthermore, throughout the free troposphere above 2 km the instruments agree quite well in their mean values of differences up to maximum $\pm 2$ ppbv. Especially above 12 km the measurements are almost identical even with respect to their standard deviations and median values.



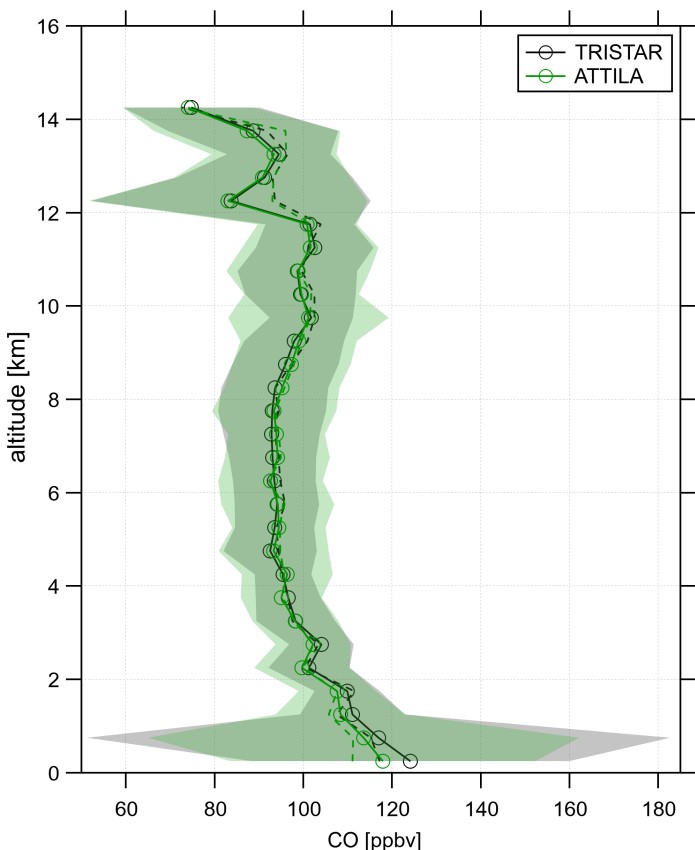

**Figure 9.** Vertical profile of all CO measurements over the whole campaign, excluding the two test flights. Both profiles are binned in 500-m boxes measured by TRISTAR in black and ATTILA in green lines. Additionally, the median is shown in dashed lines and the standard deviation of the measurements in shaded areas.

In conclusion, the ATTILA instrument is suitable for high-resolution measurements of atmospheric variations and can reliably detect atmospheric features during airborne measurements. In large part, due to its minimalist design, the data quality is

remarkable. Therefore, atmospheric spatial tracer distributions, vertical profiles and qualitative indication of potential source regions can be investigated. For example, for flux measurements which require time resolutions smaller then 1 Hz and a high data quality (Nussbaumer et al., 2023), the ATTILA instrument still has to be improved.

## 5 Results from CAFE-Brazil

With a base in Manaus, Brazil ($3°6'$ S, $-60°1'$ E), the CAFE-Brazil aircraft campaign was carried out over the Amazon

rainforest from December 2022 until the end of January 2023. Various instruments on board of the HALO aircraft measured about 50 different trace gases, radicals and aerosol particle properties in 16 local flights over the Amazon rainforest region





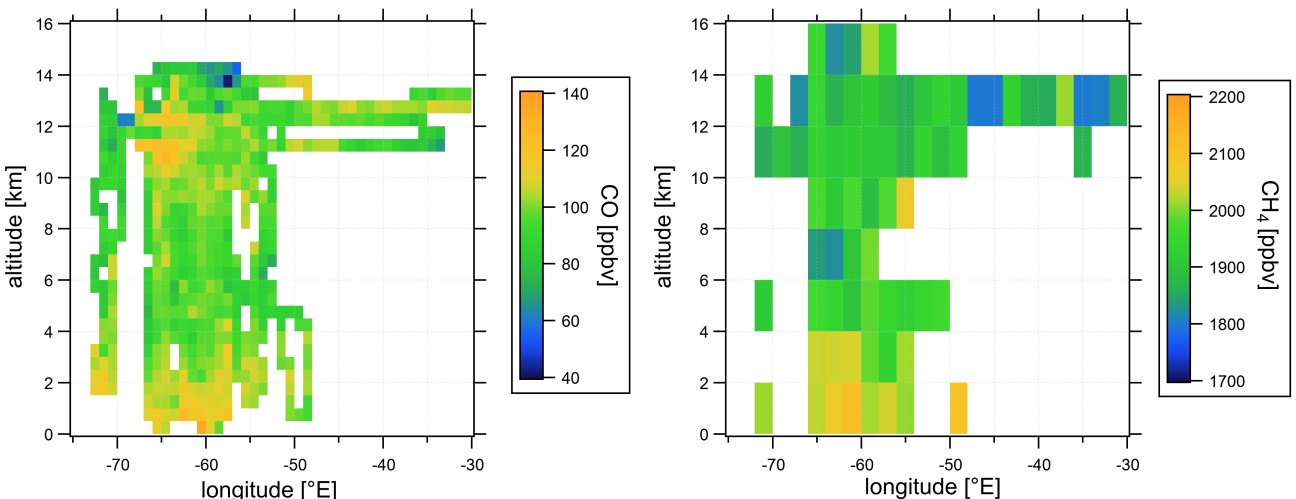

**Figure 10.** Meridional profiles of the CO (left panel) and CH$_4$ (right panel) measurements of the CAFE-Brazil campaign over the Amazon rainforest derived from ATTILA. The 1 Hz data is binned into a 1 $^\circ$ x 0.5 km subset for CO and into 2 $^\circ$ x 2 km for CH$_4$ due to differences in their $MU$ value. Additionally, for methane the bins with less than 1000 data points have been removed for a better statistical representation.

from a few meters above the forest up to about 15 km in altitude. Additionally to the local research flights, four transfer flights with a stop over in Sal, Cape Verde and two test flights over Germany have been carried out within the campaign. The main research focus lay on the investigation of oxidation processes and their control of new particle formation and particle growth over the rainforest region. Further objectives of the campaign were the investigation of the general distribution of the measured variables over the pristine forest as well as over regions influenced by anthropogenic activity and their transport through deep convection into the Tropical Tropopause Layer (TTL) (Fueglistaler et al., 2009).

In Figure 10 meridional profiles of CO (left panel) and CH$_4$ (right panel) measured by ATTILA over the Amazon rainforest from the CAFE-Brazil campaign are shown. The bin size of those distributions was chosen according to the individual $MU$s of the trace gas measurements presented in Table 1. Therefore, CO is binned into 1 $^\circ$ x 0.5 km bins and CH$_4$ into 2 $^\circ$ x 2 km bins. All data is corrected by drift, cell pressure variations and by Lambert-Beer. The CH$_4$ data is additionally corrected according to the water dilution effect, which is described in more detail in the study of Haracono et al. (2015).

Focusing on CO in the left panel of Figure 10, we can see three regions of enhanced as well as two regions of lower CO concentrations. The enhanced values originate from anthropogenic emissions and biomass burning events which can be vertically transported by convection. Specifically, in the lower altitudes below approx. 3 km, in the boundary layer, mean values reach 140 ppbv. Mostly, these are located around -60 $^\circ$E, which indicates anthropogenic emissions of the city Manaus (3 $^\circ$ 6 ′ S, -60 $^\circ$ 1 ′ E) as well as extreme single biomass burning events, which have been captured around 4 $^\circ$ 5 ′ S, -62 $^\circ$ 5 ′ E. Those elevated concentrations are vertically transported from the boundary layer into the free troposphere through deep convection and could be probed in the outflow of those convective cells several times between 10 and 13 km mostly west of Manaus. During one



research flight towards the east, another strong outflow event could be measured up to 14 km.

The lower concentrations are influenced by stratospheric air which gets transported and mixed downwards into the TTL. Two regions which are dominated by stronger stratospheric influence during the campaign are between -55 °E and -60 °E above 13 km and around -69 °E down to 12 km. That influence propagates quite deep into the free troposphere. This might be affected by gravity waves in the lee of the Andes but hasn't been investigated further in this study.

The mean concentration of $CH_4$ over the whole meridional profile above the Amazon rainforest is 1932 ppbv. According to the NOAA data base the monthly mean values for December 2022 and January 2023 are 1924.72 ppbv and 1922.16 ppbv, respectively (Lan et al., 2023). With an accuracy of 5.5 % (Table 1) for the $CH_4$ data and taking into account that the NOAA mean value has been measured over the whole vertical atmospheric column above remote marine surface these values are in quite good agreement. Nevertheless, the vertical profile of $CH_4$ also shows a different vertical distribution. Below 4 km the

mean value is 2030 ppbv and above it is 1907 ppbv. Hence, there might be a source of $CH_4$ over the Amazon rainforest region. According to the global data base from NOAA (Lan et al., 2023) there is a drastically increasing trend in atmospheric methane in the recent years. A modelling study by Zhang et al. (2023) using a wetland model, referring to the years 2000 to 2021, reports large increases in $CH_4$ emissions since 2000 with the strongest increase in the years 2020 and 2021. Based on the satellite-borne NOAA data base, Peng et al. (2022) suspected higher natural methane emissions while anthropogenic emis-

sions probably decreased over the COVID-19 lockdown. The positive feedback of natural methane emissions is suggested to be mostly from wetlands as warming and more moisture through climate change can contribute to this effect. However, as the CAFE-Brazil campaign generated a statistically representative and unique data set of in-situ measurements over the Amazon rainforest, those increased $CH_4$ values will be further investigated in another study.

# 6 Conclusions

In this study, a newly developed quantum cascade laser absorption spectrometer for airborne trace gas measurements, named ATTILA, has been presented and its functional properties and technical setup have been described. The main advantages of the instrumental design are its low weight, size and costs and its simplicity, which makes it an ideal airborne trace gas IR spectrometer and possibly attractive for commercial use. This instrument features two cells, which allows simultaneous measurements

of a minimum of two tracers. Based on a test flight from a recent airborne research campaign, CAFE-Brazil, the influence of extreme environmental changes on the signal could be identified. By identifying these influences, post-processing of the signal allows for improvement of the data quality. Therefore, temperature drift behaviour, linear pressure changes and short term fluctuations can be corrected to some extent through frequent calibrations, line-shifting and appropriate averaging of the data. Comparison with a similar, more established IR spectrometer, TRISTAR, has been used to verify the overall assumption

of the quality of the data set. It has been shown that despite ATTILA's minimalist design, the measurements are of high quality, especially at constant flight levels and high altitudes, as its main shortcomings are related to abrupt changes in altitude and extreme aircraft movements. Therefore, the compact IR spectrometer ATTILA is capable of providing detailed information about

spatial distributions in the atmosphere. The data set from the CAFE-Brazil campaign has been briefly presented and discussed and atmospheric features could be derived in the CO and $CH_4$ measurements over the Amazon rainforest. Especially for CO, only minor limitations of the data quality could be identified and detailed atmospheric variations driven by dynamical processes could be located. For $CH_4$, there are more limitations than for the CO measurements due to the optical setup. Nevertheless, a general overview of the $CH_4$ vertical distribution over the Amazon rainforest with enhanced concentrations in lower altitudes could be shown.

Through more field experience, ATTILA can be further improved. The main limitations are the influences of abrupt aircraft movements which cause non-linear etalon structures. These drawbacks can be addressed by more precise adjustment and improvement of the optical setup. With a more stable optical adjustment, the influences on the signal caused by fluctuating temperature and pressure or strong aircraft movements will be further reduced.

*Code and data availability.* All data measured during the CAFE-Brazil campaign and codes which have been used are available upon request.

*Author contributions.* HF and LO designed the study; JL and HF planned the campaign; LO and LR performed the measurements; LO and LR processed and analyzed the data; UP, RK, DC, FK, RW and HF designed the instrument; UP, RK, DC, LR and LO optimized the instrument; LO wrote the manuscript draft with contributions of all co-authors

*Competing interests.* The authors declare that they have no conflicts of interest.

*Acknowledgements.* The authors gratefully acknowledge the whole CAFE-Brazil team, including the atmospheric department of the University of Frankfurt, Karlsruhe Institute of Technology and the Deutsches Zentrum für Luft- und Raumfahrt (DLR) in Oberpfaffenhofen, whose support was essential for the project.



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
