# Peer review of "In-flight characterization of a compact airborne quantum cascade laser absorption spectrometer"

_EGUsphere, 2023_

## Author Response (AR1)

**Colorcoding:**

**Referee comments**

**Authors answers**

**Changes in text**

**CC1: J. Barry McManus, 06 Feb 2024**

Review for: Ort, et al., "In-flight characterization of a compact airborne quantum cascade laser absorption spectrometer".

General comments: This is a well written and comprehensive description of the group's new mid-IR laser spectrometer trace gas instrument for airborne measurements, named "ATTILA", including instrument design, data processing and scientific results.

Thank you for reading and commenting on our manuscript!

The figures' text is too small and lines are too thin, making them difficult to read.

We improved the features of the figures, so that they are easier to read.

Line 122: As is often argued, a presented advantage of WMS is that it is baseline independent. However, in discussion of results shown in Figure 5 [page 9], when the baseline contains optical interference fringes with free spectral range close to the frequency width of absorption lines, then those elements of the baseline are quite important. Some might say that fringes are not part of the laser baseline, but even if so in a narrow sense, fringes still may influence the measurement.

We apologize for the confusion. In WMS the baseline contribution due to laser intensity and background broadened absorption lines from open beam path absorption are greatly reduced. Further disturbances of the signal due to fringes still persist in WMS and are the main limitation of the described instrument, as discussed in the article.

Also, in discussion of results shown in Figure 5 [page 9], do the authors understand the source of the optical etalons, beyond what changes their phase?

Optical etalon structures are caused by interferences through any reflecting surfaces like mirrors and lenses. Their periodic frequencies, amplitudes and bandwidths can vary depending on their optical source in the setup. The optical source has not been determined with respect to their free spectral range since we have to deal with more than one etalon. Changes in temperature and pressure can cause small variations of the etalons free spectral range and their phase. This hardly can be identified.

I did not see a mention of how the frequency scale of the laser scan is determined. Is that in the supplemental material?

The frequency scale of the laser scan is determined by laser temperature and current according to the laser specifications and double checked by identification of the neighboring absorption lines. The frequency is kept stable throughout the measuring process.

What are the main sources of noise in the instrument?

The main sources of noise are optical fringe interferences, as discussed in the article. Further sources are electrical noise and pressure fluctuations. High frequency electrical noise caused by the detector is present but reduced by averaging the oversampled signal, as you can find in Line 113 in the manuscript.

Minor comments on the text:

Abstract, lines 15-16:  With data uncertainties and accuracies given as percentages, it is not immediately clear what are the base amounts used to calculate the percentages.

The measurement uncertainties are given relative to the calibration gas mixing ratio. The calibration gas has a CO mixing ratio of 153 ppbv and a CH4 mixing ratio of 1990 ppbv which is close to the expected ambient concentrations. We have chosen to present our measurement uncertainties and accuracies in percentage as this covers a broader range of the data. Especially, CO has a wide range in the atmosphere and thus the absolute values of the error differ depending on the absolute values. We have added this information in the abstract.

Line 15: First dynamical characteristics and tracer distributions of CO and methane (CH4) over the Amazon rainforest can be identified with ATTILA measurements with a total measurement uncertainty of 10.1 % and 17.5 % **for calibration gas mixing ratios of 153 ppbv and 1990 ppbv** and an accuracy of the standards of 0.3 % and 5.5 % for a data acquisition frequency of 1 Hz for CO and CH4, respectively.

Line 52:  Linear dimensions are given as $cm^3$.

We have acknowledged this comment and have changed it in the text.

Line 52: ATTILA is mounted in a 19-inch box and is small in size **(48 cm x 27 cm x 55 cm)** with a total weight of only 20.6 kg (excluding the pump).

Line 112:  "Sinus" wave instead of sine or sinusoidal.

We have corrected this mistake.

Line 112: With a **sine wave** modulation frequency of 17.86 kHz, a ramp is generated with 4096 discrete steps and scanned within 28.7 ms.

Line 183:  Rather than say [≈] material expansion induces etalons, perhaps:  material expansion and air pressure changes phase modulate etalon fringes that are present in the optical system.  That is, unless the authors mean that material thermal expansion produces misalignment that then produces etalons.  That would be a worse kind of problem.

We are sorry that we have not been precise enough. The cell itself is quite robust against temperature and pressure changes. Nevertheless, the surrounding optics, which lead the laser beam into the cell can be influenced by temperature and pressure changes as there is no protection mounted. This causes a phase shift of the etalon fringes while the free spectral range stays almost constant. Those multiple etalon fringes are difficult to subtract and are our main limitation of the instrument setup, as discussed in the text.

**CC2 & RC2: Jingsong Li, 13 Feb 2024 & referee #2 02 Mar 2024**

This is a very nice work. I have a few comments listed below

Thank you for reading and commenting on our manuscript!

1) The exact concentrations of the compressed air are characterized with a primary standardized gas bottle beforehand. What is the accuracy? Please specify this error.

This error is included in the calculation of Acc, which is calculated with equation 3). We have added the exact value for Ferr of the primary standard in the text.

Line 245: Another measure for data comparison is the accuracy Acc, representing the standard error of the in-flight calibration gas. Acc of the secondary standard used as calibration gas during the flights is calculated via

$$Acc = \sqrt{(RF)2 + (\sigma N)2 + (Ferr/Fmean)2}$$

including the reproducibility of the primary standard measurements RF, the standard deviation divided by the number of points of the secondary standard measurements and the primary standard error over its mean content. **The primary standard errors are given by the manufacturer (Luxfer Gas Cylinders Ltd., Colwick, Nottingham, England) with 0.36 ppbv, 0.21 ppbv and 0.24 ppbv for CO, N2O and CH4, respectively.**

2) It can be inferred from Figure 2 that the influence of humidity is not considered during the gas sampling process. How to ensure that the humidity of calibration gas is the same as the air sample gas?

The referee is correct. We do not consider the influence of humidity during the gas sampling process. As we know from Deng et al., 2017 there are spectral broadening effects from water vapor. Nevertheless, we have chosen the spectral lines and low pressure to minimize the effect of the water absorption interference. Furthermore, in the post processing we do correct our CH4 and N2O mixing ratios for the water dilution effect, which has a more significant effect on the signals depending on the water vapor content in the sampled gas. This, of course, has only an influence on the mixing ratios in the lower troposphere. In the free troposphere and higher, this effect can be neglected due to the small water mixing ratios present. For CO we aim for using actual wet concentrations to enable calculations of concentration change of reactants and products due to chemical reactions in the atmosphere.

Citation: Deng, H., Sun, J., Liu, N., Wang, H., Yu, B., & Li, J. (2017). Impact of H2O broadening effect on atmospheric CO and N2O detection near 4.57 µm. Journal of Molecular Spectroscopy, 331, 34-43.

3) The two QCLs (Alpes, Lasers, HHL-series, Lausanne, Switzerland) can measure CO (2190.02 cm−1) and N2O (2190.35 cm−1) in one cell.......... I think that the positions of the selected molecular absorption lines are not precisely given. If the related parameters are similar to previous publication (Sensors and Actuators B 182 (2013) 659- 667), it can be cited for clear comparison.

You are right. The precise line positions are at 2190.0175 cm^-1 and 2190.3498 cm^-1, same as used in the paper you mentioned. We have clarified this in the text.

Line 66: The two QCLs (Alpes, Lasers, HHL-series, Lausanne, Switzerland) can measure CO (**2190.0175** cm−1, Li et al. (2015)) and N2O (**2190.3498** cm−1, Toth (2004)) in one cell **(Li et al. (2013))** and CH4 (**1256.602** cm−1, Ba et al. (2013)) and N2O (**1255.4238** cm−1, Toth (2004)) in the second cell.

Li, J., Parchatka, U., & Fischer, H. (2013). Development of field-deployable QCL sensor for simultaneous detection of ambient N2O and CO. Sensors and Actuators B: Chemical, 182, 659-667.

4) The triangular ramp provides two spectra, an upwards- and downwards-directed spectrum of the measured signal. Please provide the frequency of the triangular wave used in this work.

We have added this information in the text in line 111.

Line 111: The triangular ramp provides two spectra, an upwards- and downwards-directed spectrum of the measured signal. With a sine wave modulation frequency of 17.86 kHz, a ramp is generated with 4096 discrete steps and scanned within 28.7 ms **in each direction. Therefore, the frequency of the triangular wave is 17.4 Hz.**

5) Here, a digital lock-in amplifier multiplies and phase 120 shifts the signal with the doubled initial modulation frequency (2f) and to match the phase of the measured signal. Please state clearly whether it is hardware lock-in amplifier or lock-in amplifier.

After the ADC the signal is further processed in the LabView program, which contains also a digital lock-in amplifier. We have clarified this in the manuscript.

Line 119: Here, a digital lock-in amplifier **demodulates** and phase shifts the signal with the doubled initial modulation frequency (2f) and to match the phase of the measured signal. **Both, the modulation and demodulation are performed by a LabView (Laboratory Virtual Instrument Engineering Workbench) controlled real-time FPGA system that is free of any jitter caused by software processes.**

6) In Figure 6: The tracer concentrations of CO in red, N2O in blue and CH4 in green in the top panel of the graph are not yet corrected by pressure and Lambert-Beer. From this figure, it can be seen that the pressure fluctuates significantly around 13:40. Theoretically, the calculated concentration values should be affected by the pressure change before correction, why the concentration data have no similar fluctuations?

The fluctuation of the cell pressure at 13:40 is about 1 hPa, which is a fluctuation of 2 %. The effects on the signal are low compared to other disturbances in the signal. Nevertheless, they can be identified and corrected for the final data over linear regression.

**RC1: Anonymous Referee #1, 22 Feb 2024**

This manuscript describes a new sensor built for airborne measurement of in situ CO, CH4, and N2O, as well as test flight and intercomparison campaign on the HALO aircraft. The sensor measures the gas concentrations using QCLs in an astigmatic multipass cell and uses WMS to improve sensitivity. The authors discuss instrument stability, intercomparison with another trace gas instrument on the same payload, and optimization of various instrument parameters.

We thank the referee for reading and commenting on our manuscript and his or her helpful comments!

My major concern with the manuscript is one of novelty. The size and complexity of the sensor is comparable to the current commercial state-of-the-art, but the performance is much worse. Based on the variability during the calibration sensitivity test flight, while the CO measurement sensitivity is likely high enough to be scientifically useful at 10 ppb effective precision, CH4 variability was reported as almost 200 ppb, which is larger than signals in all but the largest source areas. N2O was even higher, with 15 ppb variability compared to signals typically on the order of single ppb or less (but this is perhaps not actually intended to be an ambient measurement?). Time response was also low, which the authors acknowledge, but this is a major disadvantage with airborne measurements. The technique is a relatively straightforward application of WMS spectroscopy using COTS parts in manners that have been widely reported, which is not bad, but I also do not see as novel.

To make this publishable, the authors need to demonstrate the novelty of the instrument, with a particular focus on how the diminished performance is scientifically useful.

We regret that we haven't been precise enough. The novelty of this work lies in the in-flight characterization of a newly developed instrument. There has barely been a work showing the performance of a new QCLS on board of a challenging platform like an aircraft and identifying the prohibiting influences on the signal in such detail (Roller et al., 2006). From the work of Röder et al., 2023 we know that instruments can behave completely different in the laboratory compared to in-flight measurements. For a better visualization of the influence of changing platforms, we have added a comparison of two constant gas measurements of ATTILA in the supplemental material attached (Fig. S1, S2) and the manuscript supplemental material. The time series shows normalized measurements of CO and CH4 inside the laboratory and inside the HALO aircraft during the test flight, which is fully shown in the manuscript. Both time series are normalized on their standard bottle mixing ratio for a better comparison and cut into parts of equal length. The CO and CH4 mixing ratios of both standards used are for CO at 160 ppbv and 245 ppbv and for CH4 at 1920 and 2024 ppbv for the flight and lab measurements, respectively. With the same setup but in a stable laboratory our instrument measures constant timelines with measurement uncertainties of 1.6 ppb (0.67 %) for CO and 42.52 ppb (2.1 %) for CH4. Moving the instrument into the flying aircraft where the environment is influenced by changing outside conditions and aircraft movements the uncertainty increases to 8.2 ppb (5.3 %) and 195 ppb (10.2 %) for CO and CH4 for this selected interval. Hence, the changed environment changes the interferences influencing the signal, which are hard to identify. We can also notice a period during the flight, which is more stable than other parts of the timeline. Here, the aircraft flies straight on a constant level. Therefore, the characterization of an instrument during different flight pattern is crucial for understanding the data and correctly estimating the uncertainties and limitations of the instrument. With this manuscript we have not only presented a newly built instrument, we have also characterized and identified these in-flight influences and proven that this small and simple design can compete with long-time experienced instrumentation on board of such an extremely challenging platform.

Citation: Roller, C., Fried, A., Walega, J., Weibring, P., & Tittel, F. (2006). Advances in hardware, system diagnostics software, and acquisition procedures for high performance airborne tunable diode laser measurements of formaldehyde. Applied Physics B, 82, 247-264.

Citation: Röder, L. L., Ort, L. M., Lelieveld, J., & Fischer, H. (2023). Determination of Temporal Stability and Instrument Performance of an airborne QCLAS via Allan-Werle-plots.

Line 26: **Through the work of Röder et al., 2023 we also know that on airborne platform instrumental characteristics behave differently compared to the laboratory.**

Line 331: **In this study, a newly developed quantum cascade laser absorption spectrometer, named ATTILA, has been presented and intensively characterized in its in-flight measurement performance. This work not only describes the instruments functional properties and technical setup, it identifies possible data interferences for airborne trace gas measurements.**

Other comments:

Line 52: This does not seem like a low size sensor, and at 20 kg is approaching the same order as commercially available sensors.

The instrument is specifically built for airborne measurements. Of course, the size of a commercially available sensor might be smaller but not optimized for the challenging demands an airborne platform entail.

Line 66: I don't see mention of water spectral interference, which is common for these types of instruments. Is this avoided by choice of spectral lines?

The spectral lines and low cell pressure have been chosen to minimize the effect of water absorption interference. Furthermore, the interference of water absorption lines from the open beam path is negligible due to the bandpass filter effect induced by the WMS.

Line 98: is 50 hPa chosen arbitrarily or how was it decided?

The cell pressure has been chosen to be 50 hPa out of several reasons.

1) The pressure is chosen to be a good trade-off between the pressure broadening effects and a strong signal.
2) A low pressure inside the cell allows measurements high up into the atmosphere. As long as the ambient static pressure is above 50 hPa, the air flow is supported by the pressure gradient.
3) Low cell pressure reduces the effect of interferences with other absorption lines, e.g., water. High cell pressure increases the signal-to-noise ratio due to the increased concentration. 50 hPa works as a nice trade-off between these effects.
4) For a direct comparison with the TRISTAR instrument which also runs on 50 hPa cell pressure.

Line 112: sinus-wave -> sine wave or sinusoidal wave

We have corrected this in the text.

Line 112: With a **sine wave** modulation frequency of 17.86 kHz, a ramp is generated with 4096 discrete steps and scanned within 28.7 ms **in each direction**.

Line 112: I would add a brief phase or sentence mentioning the modulation depth, and how it was chosen.

We have chosen our modulation depth by finding the strongest signal-to-noise ratio of our spectral lines. This results in a modulation depth of 15 for CO and 17 for CH4. We have included a sentence in the manuscript.

Line 115: **The modulation depth for CO and CH4 have been chosen empirically to maximize the signal-to-noise ratio.**

Line 121: It's unclear what the authors mean by the 2$^{nd}$ harmonic signal "…provides a baseline near zero volts". My guess is it's meant to describe that as one tunes away from a line in wavelength, the 2F signal goes to zero, but it is somewhat ambiguous as written.

We have clarified this in the text.

Line 119: Here, a digital lock-in amplifier demodulates and phase shifts the signal with the doubled initial modulation frequency (2f) to match the phase of the measured signal. Both, the modulation and demodulation are performed by a LabView (Laboratory Virtual Instrument Engineering Workbench) controlled real-time FPGA system that is free of any jitter caused by software processes. **The 2f harmonic signal provides a nearly constant baseline in the absorption-free region and a maximum at the center of the line (Fried and Richter, 2006). This ensures a minimization of electronic noise and an in-dependency of the baseline of the spectra.**

Line 188: So, to be clear, the 50 hPa pressure in the cells is controlled by varying the pumping speed?

The cell pressure is regulated through the combination of the MFCs and the variation of the pump speed. The MFCs ensure a stable flow through the cells and the additional variation of the pump speed secures the interferences with the change of the ambient pressure by changing altitudes. As shown in Fig. 4, there is a pipe connected to the inlet line. The purpose of this line is to trap residual portions of calibration gas during the ambient measurements. Nevertheless, this line is not controlled by a MFC, hence the pump varies its pump speed to ensure stability of the cell pressure.

Figure 6: The cell pressure appeared to change a lot based on the timeseries, did I miss a mention of this being compensated for in the spectral fitting?

In the Fig. 6 you can see a timeseries from a whole test flight. During the take-off and landing phase, where turbulences, vibrations and changes of the outside pressure are most challenging for on-board instrumentation the cell pressure changes in a range of maximum 2 %. These influences cause only small variations in the signal due to the change in the number of molecules present. The effect on the spectral shape due to pressure

broadening is negligible. Nevertheless, the pressure broadening effects can be corrected afterwards, as discussed in the text.

Line 229: This variability in N2O and CH4 is very large compared to all but the largest sources? What science is this instrument aimed at investigating?

The science we are aiming to investigate are chemical reactions of the traces gases we measure. With the wet CO mixing ratios, we can identify sink processes by reactions with OH as well as transport processes through the lower atmosphere. Furthermore, according to Bozem et al., 2017, the ratio of CH4 and CO can estimate the ratio of CH3O2 and HO2, which are important reactants for ozone formation. The N2O absorption line is only used for line locking reasons to clearly identify the position of the CO line even in stratospheric conditions where CO mixing ratios decrease. It was not planned for scientific use.

Line 284: In large part, due to its **minimalist and robust design during airborne measurements, this instrument can be used for dynamical and chemical investigations of the atmosphere.** Therefore, atmospheric spatial tracer distributions, vertical profiles and qualitative indication of potential source **and sink regions through chemical reactions** can be investigated. For example, for flux measurements which require time resolutions smaller than 1 Hz and a high data quality (Nussbaumer et al., 2023), the ATTILA instrument still has to be improved.

Citation: Bozem, H., Butler, T. M., Lawrence, M. G., Harder, H., Martinez, M., Kubistin, D., ... & Fischer, H. (2017). Chemical processes related to net ozone tendencies in the free troposphere. Atmospheric Chemistry and Physics, 17(17), 10565-10582.

Line 301: Why is only methane corrected for water dilution, and what water measurement is used to do so? In my experience, is it standard practice to report both carbon monoxide and nitrous oxide in dry mixing ratio as well, especially given that calibration gases are measured in that state.

It is indeed important to correct the mixing ratios to identify detailed variations and trends of greenhouse gases. For the correction we use H2O volume mixing ratios measured by SHARC, an absorption spectrometer run by the DLR (Oberpfaffenhofen, Germany). Furthermore, the water dilution is only important in lower altitudes where the water mixing ratio is high. In the free troposphere and higher this effect is neglectable due to the decreasing water vapor concentrations. However, for the science this instrument is aiming for lies in the understanding of the chemical composition and reactions in the atmosphere. Wet mixing ratios enable calculations of concentration change of reactants and products due to chemical reactions. However, the N2O measurements are only used for line locking purposes for the CO spectra as in the stratosphere the CO mixing ratios are close to the detection limit while N2O ensures the right position of the spectra.

Line 315: These background methane numbers are relatively representative of the global background, and with minimal methane sinks in the troposphere, how do you explain the very low concentrations observed (below 1800 ppb?). Those levels can be observed through stratospheric influence, but one would expect similar CO decreases.

Thank you for your comment. The meridional distributions show the mean values of CO and CH4 over different sized boxes from the whole CAFE Brazil campaign data set. As mentioned in the text, the tracers and hence the two cells show differences in their uncertainties. CH4 is more effected by undefinable etalon structures which produce sometimes outliners which can influence the averaged distribution. However, we have changed the meridional distributions to the median values in the manuscript as those are not that effected by those outliners.

General:

- What happened to discussion of the N2O channel? This is described as a functional measurement, but on line 240, the authors state that it is only used as a reference signal?

We regret the confusion. The N2O channel is fully functional, but its main purposes are providing a line position reference when CO concentrations are very low and an identification of instrumental artifacts. For atmospheric data analysis the measurement performance of N2O is (still) too low. N2O varies quite little in the troposphere and hence small variations in the ambient measurements cannot be identified. Nevertheless, as we approach the tropopause region and enter the stratosphere, we do see decreasing N2O signals, as we would expect. But, the purpose of this work lies in the in-flight characterization of the instrument. The optical adjustment, as pointed out in the outlook in line 350, can be, of course, further improved which would ensure more detailed features of all species measured.

- There is no timeseries of ambient data of any of the species, which makes it difficult to evaluate artifact sensitivity once the data is calibrated.

We are sorry for this confusion. In Fig. 7 in the manuscript, you can find a time series of CO ambient measurements. We decided that including time series of ambient data would deviate from the main purpose of the article and lead to confusions. Instead, we decided to show the meridional distributions of the tracers, as they show a better overview of the whole field campaign than a single flight and emphasize the purpose of science which is aimed to be investigated. Nevertheless, we have added a time series of an example flight (RF19) in the supplemental material attached (Fig. S3) and in the supplemental material of the manuscript to show the in-flight comparison of fully processed CO data measured by ATTILA and TRISTAR. During this flight the MU for the CO measurements was at 7.8 % and 2.5% for ATTILA and TRISTAR, respectively.

[Figure]

**Figure S1.** Two time series of constant gas measurements of CO while the ATTILA instrument was built in the rack and located in the laboratory (yellow) and inside the aircraft HALO during the test flight on the 22$^{nd}$ of November 2022 (red). Both time series have been normalized to the standard concentration of the gas bottles used for a better comparison. The concentrations of the standard gas bottles are at 156 ppbv and 245 ppbv for the flight and laboratory, respectively. The measurement uncertainties for the laboratory CO measurements are 1.6 ppbv (0.665 %) and for the aircraft on ground measurement at 8.2 ppbv (5.3 %).

[Figure]

**Figure S2.** Two time series of constant gas measurements of $CH_4$ while the ATTILA instrument was built in the rack and located in the laboratory (yellow) and inside the aircraft HALO during the test flight on the 22nd of November 2022 (red). Both time series have been normalized to the standard concentration of the gas bottles used for a better comparison. The concentrations of the standard gas bottles are at 1920 ppbv and 2024 ppbv for the flight and laboratory, respectively. The measurement uncertainties for the laboratory $CH_4$ measurements are 42.52 ppbv (2.1 %) and for the aircraft measurement at 195 ppbv (10.2 %) for the shown period.

[Figure]

**Figure S3.** An example research flight (RF19) of fully processed ambient CO data during the CAFE Brazil campaign is shown. The CO mixing ratios of ATTILA are shown in red and from TRISTAR are shown in blue. Additionally, the GPS altitude given by the BAHAMAS instrument (DLR, Oberpfaffenhofen, Germany) is displayed in black. The measurement uncertainty of this flight was at 2.5 % and 7.8 % for TRISTAR and ATTILA CO measurements, respectively.